# The Influences of Oral Probiotics on the Immunometabolic Response During Pregnancy and Lactation: A Systematic Review

**DOI:** 10.3390/nu17091535

**Published:** 2025-04-30

**Authors:** Valentin Nicolae Varlas, Laurențiu-Camil Bohîlțea, Nicolae Suciu

**Affiliations:** 1Department of Obstetrics and Gynecology, Carol Davila University of Medicine and Pharmacy, 020021 Bucharest, Romania; 2Department of Obstetrics and Gynecology, Filantropia Clinical Hospital, 011132 Bucharest, Romania; 3Department of Medical Genetics, “Carol Davila” University of Medicine and Pharmacy, 050474 Bucharest, Romania; 4Fetal Medicine Excellence Research Center, Alessandrescu-Rusescu National Institute for Mother and Child Health, 020395 Bucharest, Romania; nicolae.suciu@umfcd.ro; 5Department of Obstetrics and Gynecology, Alessandrescu-Rusescu National Institute for Mother and Child Health, Polizu Clinical Hospital, 020395 Bucharest, Romania

**Keywords:** probiotics, pregnancy, lactation, immunometabolic response, glucose metabolism, gestational diabetes mellitus, overweight/obesity, preeclampsia, allergy, eczema, atopic dermatitis

## Abstract

Background/Objectives: In recent years, due to the emergence of antimicrobial resistance, probiotics have been increasingly used during pregnancy and lactation with real maternal–fetal benefits. Probiotic intervention, especially multi-strain probiotics, due to their anti-inflammatory, metabolic, and immunomodulatory actions, can be performed prophylactically and therapeutically with promising results regarding maternal, fetal, and neonatal health. The administration of probiotics can modulate the maternal microbiome, regulate microflora imbalance in various conditions (overweight/obesity, gestational diabetes mellitus (GDM), preeclampsia, allergic diseases), and influence several reactions such as modulating the non-specific cellular immune system, metabolic processes, and inhibition of pathogens. This study aimed to analyze, based on available data, how the administration of probiotic supplements to women during pregnancy can modify immunometabolic responses to microbial dysbiosis to limit weight gain and the risk of obesity, to improve glucose homeostasis and reduce the risk of GDM, to prevent preeclampsia and its effects on maternal–fetal outcomes, and to reduce rates of atopic eczema and allergic diseases in infants. Methods: We performed a systematic search in MEDLINE/PubMed to identify studies that have investigated the effects of probiotic intervention on the immunometabolic response in pregnancy and lactation, especially in women with diabetes, overweight/obesity, preeclampsia, and allergic conditions. Results: Fifty-six RCT studies, totaling 15,044 women, matched the inclusion criteria, of which eight were for interventions on the immune response, twenty on allergic conditions, seven on obesity and excess weight gain in pregnancy, and twenty-one on GDM. Conclusions: Due to the heterogeneous structure and the size of the samples, the methodologies, formulations, moment of initiation, and study durations, future research is needed to establish their effectiveness and safety in pregnancy and lactation regarding maternal-fetal health and outcomes in childhood and adult life.

## 1. Introduction

The use of probiotics during pregnancy is continuously increasing due to the multiple benefits they provide to pregnant women’s metabolic processes and immune systems. An essential role regarding maternal-fetal health is represented by the interactions between the different components of the body’s microbiota (oral, intestinal, vaginal, possibly placental), which define a whole group of microorganisms with an active role in development processes and immune defense responses against pathogens [1].

Maturation of the fetal immune system is initiated during pregnancy and continues after birth, through the presence of immunogenic stimuli, especially from the intestinal microbiota. Microorganisms that are part of the intestinal microbiota constellation are found in the pregnant uterus, identified in the placenta, meconium, and amniotic fluid. As a result, the myth that the uterine environment is sterile is disproved by some species of microorganisms from the genera *Lactobacilli*, *Staphylococcus*, and *Bifidobacterium* [2].

An imbalance of the maternal gut microbiota, amplified by factors such as diet, genetics, medications, parity, weight gain, smoking, and sepsis, can cause complications during pregnancy (diabetes mellitus, preeclampsia), secondary to immunological changes at the maternal-fetal interface. Prenatal initiation of the immune system is achieved by the adhesion and crossing of the intestinal epithelial barrier of microbial metabolites or by the release of mediators such as short-chain fatty acids (acetate, propionate, and butyrate), lipopolysaccharides, or extracellular vesicles. Changes in the gut microbiome in pregnant and non-pregnant women can affect immunometabolic processes, with downstream effects on various organs and tissues, leading to significant consequences for maternal and newborn health. As a result, targeted action on the gut microbiome may be a method for preventing pregnancy-related diseases [3].

The mode of delivery, diet, comorbidities, and external factors influence the early development of the infant’s microbiota. Host–microbiota interactions in pregnancy directly affect the metabolic and immunological responses that allow the development of the fetal allograft and protect it from various external factors or pathogens. This process is based on a balance between inflammatory processes and anti-inflammatory factors, requiring adaptation and modulation of the immune response at each stage of pregnancy development [4]. Mor et al. [5] described a specific pattern of these processes: implantation and placentation being favored by the presence of an inflammatory environment; the tolerance and development of the fetal allograft require an anti-inflammatory environment so that in the end, birth is favored by increased pro-inflammatory levels.

During pregnancy, the immune system must maintain tolerance to the fetal allograft and adapt immune mechanisms against pathogens. An increased rate of complications such as spontaneous abortion, preterm birth, gestational diabetes, and preeclampsia accompanies the dysfunction of these processes [6]. To increase the efficiency of probiotics, we must analyze the type of stem/stems used, the amount of microorganisms, the formulation, the administration path, and the duration of the intervention. The consumption of probiotics during pregnancy and/or lactation containing *Lactobacilli*, *Bifidobacterium*, *Propionibacterium freudenreichii* subsp. *shermanii* JS, and *Streptococcus thermophilus* STY-31 induce beneficial changes to the microbiota of newborns. As a result, multiple therapies have been developed that combine several common species and personalized formulas containing nanoparticles [2,4] or probiotics [5,6].

This article aims to highlight the interrelationships between probiotics and the maternal microbiome and metabolic pathways, synthesize the prophylactic and therapeutic effects of their interventions on perinatal outcomes, and discuss the potential benefits in various conditions (gestational diabetes, preeclampsia, allergies, and overweight/obesity).

## 2. Materials and Methods

### 2.1. Overview

We systematically reviewed the scientific literature on the selected topic in the present study using the Prisma (Preferred Reporting Items for Systematic Review and Meta-Analysis) methodology [7]. The investigated groups were pregnant women and their newborns; the studies compared the administration of probiotics to mothers, compared with a control group. The analyzed outcomes were related to the risk of diabetes, overweight/obesity, preeclampsia, allergy, or atopic dermatitis.

### 2.2. Database Sources and Electronic Search Strategy

In this review, we aimed to evaluate and synthesize studies from the literature to identify correlations between immunometabolic processes and the administration of probiotics during pregnancy and lactation. To conduct this review, we searched one database (MEDLINE/PubMed) for articles written in English, covering all available years until 25 May 2024. For this purpose, according to the PICO recommendations, the following MeSH keywords were used: “probiotics”, “pregnancy”, “lactation”, “intestinal microbiota”, “metabolic syndrome”, “diabetes”, “obesity”, “overweight”, “preeclampsia”, “preterm birth”, “periodontitis”, and “immune system”.

### 2.3. Study Design

The selection of articles included in the study was based on the relevance of randomized clinical studies, and the full texts that were included the subsequent analysis were based on the size of the samples, methodology, type of form, moment of initiation, duration of the study, and statistical significance. The inclusion criteria were the administration of probiotics during pregnancy and lactation, with outcomes related to metabolic syndrome, immune system, diabetes, preeclampsia, allergies, and atopic dermatitis. Studies were excluded from this study if they did not meet these criteria, as well as lacked data regarding the effect of probiotics, or used probiotics in combination with prebiotics, synbiotics, or other types of intervention.

### 2.4. Data Extraction, Analysis, and Assessment of Quality and Risk of Bias

The evaluation of data from the extracted studies were analyzed independently by two reviewers, who processed the titles and full texts of the eligible articles. The analyzed information included the design of the study, dimensions of the study and control groups, type of probiotic or the combination of probiotics, method of formulation, duration of treatment, timing of intervention, obtained results, and conclusions regarding their degree of efficiency. To evaluate the risk of bias across different domains, the Cochrane Risk of Bias assessment tool was used, allowing the identification of the study’s low, medium, or high risk of bias [8].

## 3. Results

### 3.1. Search Results

We identified 849 records (Figure 1) after the initial manual search using the planned keywords. After further analysis, 293 duplicates were retrieved among them. Following screening by title and abstract, 124 records remained. Finally, we analyzed 56 full-text articles that met the eligibility criteria.

### 3.2. Main Characteristics of the Selected Studies

Fifty-six studies were included in this review, of which eight focused on intervention on the immunometabolic response (Table 1), twenty on allergic conditions (Table 2), seven on overweight/obesity in pregnancy (Table 3), and twenty-one on GDM (Table 4).

## 4. The Maternal Microbiome—The Key to Immunometabolic Responses and Influence on Infant Microbiota

The gut microbiota composition in non-pregnant women is predominantly dominated by *Bacteroides* and *Firmicutes* species, and to a lower degree by *Proteobacteria*, *Fusobacteria*, *Actinobacteria*, and *Verrucomicrobia* species. The gut microbiota in the first trimester of pregnancy is similar to that of non-pregnant women, with the difference that in early pregnancy, it is dominated by *Firmicutes* (*Faecalibacterium prausnitzii*, *Clostridiales*) relative to *Bacteroidetes*. On the other hand, in pregnant women’s microbiota in the third trimester, there is an increase in the species density of *Lactobacilli*, *Proteobacteria (Enterobacteriaceae*, *Escherichia coli*), and *Actinobacteria* phyla (*Propionibacterium*), and a reduction in alpha diversity, *Faecalibacterium prausnitzii*, and *Roseburia intestinalis* species [65].

Some studies have not shown changes in the overall gut microbiota of pregnant women compared with non-pregnant women, with differences being determined by ethnicity and region [66,67]. The taxonomic composition characterized by enterotypes did not undergo significant changes independent of gestational age, with a small reduction in the Ruminococcus phylum in the third trimester of pregnancy [66].

However, Koren et al. [68] highlighted that in the third trimester of pregnancy, an increased density of intestinal *Proteobacteria* was associated with increased levels of IL-2, IL-6, IFN-γ, and TNF-α, demonstrating a low inflammatory level. The mechanism by which probiotics act on the maternal intestinal microbiota, which can influence the placental metabolome, is unknown [69]. (Figure 2) In the last two decades, the intestinal microbiota of pregnant women has undergone a progressive decrease in richness and biodiversity, reaching a composition similar to that of overweight women [70].

The microbiome of infants delivered by cesarean section, determined by microorganisms from the environment, is different from that of those delivered vaginally, the latter being dependent on colonization with vaginal fluids and maternal feces. Differences in the intestinal microbiota depending on the birth route are significant in the first three months of life and disappear after 6 months, when the immune system matures. The fecal metabolome of those delivered vaginally revealed an increased density of *Bifidobacterium*, *Lactobacilli*, *Actinobacteria*, *Bacteroides*, and *Parabacteroides*, and a higher metabolic rate of tryptophan and pyruvate compared with infants delivered by cesarean section, who showed an abundance of *Klebsiella*, the phylum *Firmicutes*, and increased activity of ABC transporters [71,72]. Furthermore, the microbiome of infants delivered by cesarean section contains increased levels of *Enterococcus*, which is associated with changes in immune cells, and decreased levels of *Bacteroidetes*, which are associated with an increased risk of obesity and type 2 diabetes [73].

Immediate intervention after birth on the intestinal microbiota from the cesarean section favors *Bifidobacterium* colonization, bringing it closer to the microbiota of newborns from vaginal birth within a maximum of one week [1]. In a randomized clinical trial of 68 infants delivered by cesarean section, Zhou et al. [74] observed that in the group that received vaginal microbiota transfer, the maturation of the gut microbiota was accelerated within 42 days of birth, directly influencing the carbohydrate and amino acid metabolome, with partial improvement of neurodevelopment in these infants.

Pregnancy induces a pro-inflammatory status by increasing cytokine levels. As a result, a more or less pro-inflammatory maternal diet can be associated with various adverse reactions, particularly evident in obese women. A pro-inflammatory diet associated with diabetes or obesity may induce pro-inflammatory oxidative processes in the trophoblast and placenta, leading to impaired development. Thus, a chronic inflammatory response may affect cellular metabolism, with adverse effects on fetal growth and neurodevelopment, as well as telomere shortening, which plays a role in aging processes. Fetal pro-inflammatory responses transferred from the mother have been observed in the context of maternal obesity, increased insulin resistance, oxidative stress, and alterations in lipid and glucose metabolism [75].

Breastfeeding plays an essential role in the health and development of the infant by stimulating immunomodulatory responses and reducing inflammatory responses. Therefore, diet plays an important role in preventing the development of food allergies in infants, not necessarily by excluding these foods, but by using probiotic preparations to reduce the inflammatory response and increase immunological tolerance [76]. Although data are limited, during pregnancy and lactation, diets high in fat and protein may affect the microbiota, leading to the development of intestinal inflammation and ultimately disrupting the health of newborns [77].

## 5. The Changes of the Probiotic Intervention on the Immune Response During Pregnancy and Lactation

The interdependence between the complex microbiota and the cellular immune system creates homeostasis between the maternal host and microorganisms, creating a balanced environment that promotes beneficial bacteria and the defense against pathogenic bacteria. Along with the evolution and diversity of the microbial community at each compartment level (gut, oral, placental, vaginal), the immune system has progressively evolved with critical interventions regarding the developmental stages of the product of conception. Thus, the microbiota and the immune system mature in parallel, with major epigenetic effects in adult life [78].

The old paradigm regarding the sterile character of the uterus seems to be refuted by new studies, which suggest that the initial interaction with the microbiota does not occur at birth but even in utero, where intracellular bacteria without septic potential have been visualized histologically. In support of this statement are research studies in which fragments of bacterial DNA were identified in the blood of the umbilical cord, amniotic fluid, and placenta, without subclinical or clinical manifestations [79]. Although viable bacteria were not detected, it is assumed that during pregnancy, there is a translocation of elements belonging to the maternal microbiota to the fetal level [80,81]. Jiménez et al. [82] demonstrated the existence of a marked maternal-fetal passage of *Enterococcus faecium* in mice, from the gastrointestinal tract to the immune cells in the blood through the mesenteric lymph nodes.

Immune reactivity to the action of microbes in the intestinal tract is based on the partial migration of dendritic cells containing bacteria and genetic material from Peyer’s patches into the mesenteric lymphoid tissue. Afterward, bacterial components are translocated from these mononuclear cells, transported endogenously through peripheral blood or lymph, and then found in the dendritic cells of lactating breast tissue. This mechanism demonstrates the impact of bacteria-laden breast milk cells on the newborn’s immature immune system [83].

On the other hand, uterine colonization through the rise of the vaginal microbiota can cause chorioamnionitis, with an increased risk of premature birth [84]. The microbial colonization pattern of premature newborns is different from that of those born at term due to the presence of risk factors such as gestational age, mode of delivery, duration of hospitalization in the neonatal intensive care unit (NICU), days of parenteral nutrition, administration of antibiotics, mode of feeding, neonatal complications, and the type and duration of probiotic use. Therapeutic strategies must consider all these factors to achieve a healthy microbiota in these infants because deficient initial colonization can have long-term effects on their growth and development [85].

Forsberg et al. [14], in a multicenter, double-blind study (PROOM-3), which enrolled pregnant women from 20 weeks of gestation and who received supplements with *Limosilactobacillus reuteri* (formerly *Lactobacillus reuteri*) and ω-3 PUFA, demonstrated by flow cytometry the immunomodulatory effects of probiotics administrated in the second half of pregnancy on activated and regulatory T cells (Treg, CD45RA—Foxp3++/CD45RA + Foxp3+). Another additional study on allergy prophylaxis using the same probiotics highlighted transcriptional effects on neonatal T helper cells [86].

In the RCT Probiotics in the Prevention of Allergy among Children in Trondheim (ProPACT), 415 pregnant women were randomized to receive probiotics *Lacticaseibacillus rhamnosus* GG (formerly *Lactobacillus rhamnosus* GG) (LGG), *Bifidobacterium animalis* subsp. *lactis* Bb-12 (Bb-12), and *Lactobacillus acidophilus* La-5 (La-5) versus placebo, with atopic dermatitis (AD) assessed over two years of in their newborns. Investigating regulatory Th cells (Th1, Th2, Th9, Th17, and Th22), the study found a reduction in the proportion of Th22 cells in children in the probiotic group compared with placebo [13].

The preventive effect of probiotics on AD is explained by the increase levels of plasma CRP, IgE, IgA, and IL-10, and probiotic-induced low-grade inflammation, as an immune modulator protecting against allergy [10], partially by the reduction of the percentage of Th22 cells [13]. Furthermore, Chen et al. [84] observed a significant increase in IL-1β, IL-2, IL-12, and IFN-γ levels in all groups and an immunomodulatory effect of probiotic intervention, which induced an increase in the levels of the cytokines IL-5, IL-6, TNF-α, and GM-CSF, followed by a pro-inflammatory status observed in the third trimester, which prepares the maternal body for labor.

The MicrobeMom2 study analyzed the maternal immune response changes during pregnancy, influenced by parity and body mass index (BMI). Primiparous women showed higher leptin levels at the end of pregnancy, while multiparous women had lower levels of PBMC (peripheral blood mononuclear cells)-derived TNF-α, IL-10, and IFN-γ levels with gestation [87]. Thus, the diversity of the maternal microbiota during pregnancy intervenes in the immune programming of the fetus and newborn. The initial interaction with microbial diversity is likely intrauterine. The direct influence of the maternal microbial environment on infant immune development is observed in the prophylaxis of allergy (eczema) in infants with prenatal and postnatal supplementation [88].

Following the analysis of eight RCTs (Table 1) totaling 1319 patients regarding the probiotic intervention on the immune response during pregnancy and lactation, a favorable response was observed in six studies [9,10,12,13,14,15] and no response in two studies [11,16]. Probiotic intervention with *Lcb. rhamnosus* GG was observed in five studies [9,10,11,12,13] (in two studies as a single component), *Lcb. rhamnosus* HN001 in one study [15], *L. acidophilus* and *Lmb. reuteri* in two studies [13,14], and *B. longum* in one study [16]. In probiotic interventions with *Lactobacilli* sp., the efficiency was 66.6%, compared to no efficiency with *Bifidobacterium*.

## 6. The Influence of Probiotic Intervention on Allergic Conditions

The high incidence of allergic diseases, especially among infants, children, and adolescents, represents a serious health problem with a significant impact on quality of life. The evaluation of the efficacy of probiotics has not yet been fully proven, a fact demonstrated by both pro and con studies. However, probiotics can be used as a beneficial adjuvant therapy by appropriately modulating immune responses in atopic dermatitis, allergic rhinitis, and asthma. The mechanisms are multifactorial and present individual variations. Probiotics can intervene in treating allergic diseases through several mechanisms, such as suppressing the host inflammatory response by decreasing circulating cytokine levels, increasing the tolerance of immune responses, and modulating intestinal barrier function [89].

Although it is not yet known how prenatal microbial exposure can have immuno-modulatory effects in humans, the effectiveness of probiotics in the prophylactic and therapeutic intervention against eczema in infants has been demonstrated. Another role of probiotics is in mitigating the risk associated with the administration of antibiotics during pregnancy, which is associated with the development of allergic terrain (atopic dermatitis) [90,91] and childhood asthma [92]. Many studies have been conducted on rodents; however, the results cannot be extrapolated to humans because research has shown differences in the adaptive immune system. In humans, immune maturity occurs rapidly before birth, with the presence of intestinal B and T cells by 14 weeks of gestation [93,94].

Exclusive feeding of infants with powdered milk increases the incidence of infectious diseases and allergic conditions due to the immaturity of the immune system. The administration of probiotics corrects this deficiency by supplementing the immune factors necessary for developing mucosal immunity. This effect was demonstrated by the evidence of higher sIgA levels in these infants’ feces [95]. Most clinical studies show the effectiveness of probiotics in infants in the prophylaxis and treatment of diarrhea syndromes and allergies, without knowing their effectiveness among healthy infants. In a randomized, double-blind study conducted on 200 infants aged 4–6 months who received a probiotic containing *B. infantis* R0033, *B. bifidum* R0071, and *L. helveticus* R0052 daily for 4 weeks, clinical benefits regarding their health status were observed [96].

Following the analysis of 20 RCTs (Table 2) totaling 7817 patients regarding probiotic intervention on atopic diseases, a favorable response was observed in four studies [17,19,26,33] and no response in five [18,22,32,35,36]. For eczema/atopic eczema, a favorable response was reported in ten studies [18,21,23,24,26,29,30,31,32,33] and no response in four [19,25,34,35].

Prenatal/postnatal preventive administration of maternal and infant probiotics is safe and effective in reducing the risk of atopic eczema. Probiotic intervention with *Lcb. rhamnosus* HN001 from 14 to 16 weeks until birth, and continuing for 6 months postnatally during breastfeeding, reduced the prevalence of eczema and atopic disease in infants at one year of age [33,34]. Administration of a probiotic cocktail (a mixture of 3–4 probiotics, represented by *Lactobacilli* and *Bifidobacterium*), over a shorter period (from 36 weeks to 3–12 months postnatally), reduced the incidence of atopic eczema in childhood [21,23,24,30], but not of other allergic conditions (asthma, AS) [32]. Probiotics administered prenatally from the second month and 2 months postpartum have proven effective in reducing the risk of eczema without having any effect on the risk of AS in infants [29]. In conclusion, the beneficial effects of probiotics on atopic eczema in high-risk infants were observed [19,21,23].

Regarding the risk of sensitization, there was a favorable response in two studies [20,30] and no response in the other two trials [24,29]. One study demonstrated the protection of probiotics only for babies delivered by cesarean section [22]. Probiotic intervention in asthma management has not shown any benefit [32,34]; only one study suggested a potential reduction in the risk of developing a respiratory disease [19].

Probiotic interventions using *Lcb. rhamnosus* GG/HN001 were observed in 14 studies [17,18,20,22,24,25,26,27,28,29,32,33,34,36], with other strains of *Lactobacilli* in four studies (alone or in combination with *Bifidobacterium*) [19,23,30,35], and *Bifidobacterium* in two studies [21,31]. The administration of probiotics started either at the end of the first trimester [20,31,33,34,36] or after 35–36 weeks of gestation [17,18,19,21,22,23,24,25,26,27,28,29,30,32,35], and subsequently continued during infant breastfeeding.

## 7. Probiotics Intervention Improves Glucose and Lipid Metabolism in Pregnant Women

During pregnancy, changes in the intestinal microbiota produce metabolic dysfunctions with pro-inflammatory effects, increased energy consumption, and decreased sensitivity to insulin. In the case of pregnant women with a high BMI, intestinal microbiota dysfunctions can lead to gestational diabetes mellitus (GDM) [97].

In pregnancy, fasting plasma glucose (FPG) is significantly reduced compared to pregnant women with GDM where FPG did not improve. When probiotic intervention is performed in the third trimester of pregnancy, a trend toward a decrease in FPG is observed. Intervention with multiple probiotics causes a reduction in serum insulin levels and insulin resistance (HOMA-IR), without being correlated with the quantitative insulin sensitivity check index (QUICKI). Concerning glucose metabolism, probiotics from several species proved useful if the duration of the intervention was ≥8 weeks [97]. During pregnancy, the decrease in inflammatory response (reduction in IL-6, TNF-∝, and hs-CRP levels) following probiotic intervention is the mechanism that will cause significant reductions in FPG, insulin, and HOMA-IR [50]. To achieve this action, a probiotic product should contain a sufficient dose of >10^8^–10^10^ CFU/day of viable cells [98].

Probiotic interventions in overweight/obese pregnant women do not modify FPG regardless of the duration of their administration [46,54]. Obesity and changes in lipid metabolism are closely related to the increased presence of bacteria from the genera *Blautia* and *Ruminococcus*, based on changes in biotin metabolism, glycosyltransferases, and oxidative phosphorylation pathways [99]. However, the strains *Blautia luti* and *Blautia wexlerae* exert antibacterial, anti-inflammatory, and metabolic effects, having as a mechanism of action the increase in butyrate production with the control of blood sugar and inflammatory processes related to anti-obesity [100].

An increased level of triglycerides and a low level of HDL is observed in pregnant women, and the administration of probiotics could postpartum decrease the concentration of total cholesterol and triglycerides (LDL-C) [101]. During pregnancy, short-chain fatty acids (SCFA), resulting from the fermentation processes of the intestinal microflora, are detected by the free fatty acid receptor 2 (FFA2) in the intestinal tract and the peripheral blood, with a role in regulating glucose homeostasis [102]. Other roles regarding the use of probiotics during pregnancy are related to the influences on the intestinal microbiota’s metabolism of flavonoids, with roles in immune and anti-inflammatory modulation, peptidases in protein breakdown, and lipid biosynthesis proteins [100].

The intestinal microbiota axis—SCFA—glucagon-like peptide-1 (GLP-1) is a mechanism in the metabolic reactions through which probiotics influence the decrease of *Firmicutes* and the increase of *Bacteroides* and *Bifidobacterium*, followed by the elevated level of butyrate and the release of GLP-1 [103].

Another possible mechanism is the probiotic–intestinal flora–butyrate–inflammatory pathway, followed by treatment of low-grade inflammation by reducing the inflammatory markers (TNF-α and IL-6) and glucose levels, and increasing the levels of GLP-1 and insulin sensitivity in pregnant women [97]. Cathepsin D is a biomarker dependent on metabolic disorders, whose pregnancy levels have not been modified by probiotics, and the reduction of these values in the third quarter is accompanied by a low pro-inflammatory level [104]. Zheng et al. [105], in a meta-analysis of ten randomized trials, observed that in women with GDM, no correlations between the use of probiotics and the levels of lipids, total cholesterol, HDL-c, LDL-c, or triglycerides were identified, while another study identified significant reductions in total cholesterol and triglycerides after metabolic intervention with probiotics [97].

## 8. The Effects of Probiotics on Obesity and Excessive Gestational Weight Gain

The interrelation between obesity and intestinal regulation is an area of interest regarding therapeutic immuno-nutrition promoted by probiotic intervention against metabolic syndrome [106]. Maternal obesity is a widespread worldwide epidemiological problem that is accompanied by adverse maternal and neonatal outcomes. Vähämiko et al. [37] showed that the mechanism of action of probiotics on metabolic disorders caused by obesity is represented by the DNA methylation status of the genes responsible for weight gain. The prevention of metabolic syndrome through the intervention of probiotics [for example, *Lcb. rhamnosus* or *Lacticaseibacillus casei* (formerly *Lactobacillus casei*)] is achieved through the immunomodulatory effect of preventing chronic inflammatory states of low degree. This mechanism is realized by activating the pro-inflammatory cascade through the action on the TLR pathway, the degradation of IĸB kinase, and the release of nuclear factor-kappa B [107].

During gestation, obese pregnant women exhibit greater insulin resistance compared to non-obese pregnant women, higher levels of plasma insulin, IGF-1, leptin, and lower plasma adiponectin concentrations, which activate mTOR-mediated placental protein synthesis and may be associated with an increased risk of fetal macrosomia and GDM. Although maternal obesity affects placental metabolic function, most children of these women have normal birth weight, demonstrating that the mechanisms are still incompletely elucidated [108].

Preconceptionally, women with high BMI show changes in the intestinal microbiome, which will be preserved even in pregnancy, with a high content of *Bacteroidetes*, *Clostridium*, and *Staphylococcus*, and with a smaller number of *Bifidobacterium* and *Akkermansia muciniphila*. The infants of these women had an abundance of *Bacteroidetes* and *Staphylococcus* in the feces and a low level of *Bifidobacterium* compared to women with normal weight [109]. Other studies on obese pregnant women who were given probiotics did not highlight any significant difference regarding the excessive increase in weight, HbA1c, newborn weight, or the risk of GDM, although a modulation of the diversity of the intestinal microbiota was achieved [54,110,111]. Another study by Saros et al. [42] observed that administering probiotics alone or in combination with fish oil to overweight/obese women from early pregnancy to 6 months postpartum reduced the percentage of body fat in their children aged <24 months. In addition, the prenatal administration of the multi-strain probiotic Vivomixx^®^ to obese mothers was associated with a low prevalence of the obesity-associated *Collinsella* genus in the infant gut microbiota [112].

Postpartum intervention with multi-strain probiotics for 12 weeks in post-GDM women resulted in decreased FPG levels, increased *B. adolescentis* by modulating gut dysbiosis, and did not alter BMI [41], in comparison with the results reported by other studies [49,57].

Following the analysis of seven RCTs (Table 3) totaling 1872 patients regarding the probiotic intervention on obesity and excessive weight gain, a possible favorable response was observed in four studies [37,40,41,42] and no response in three studies [38,39,43]. The probiotic intervention with *Lcb. rhamnosus GG* was identified in two studies [37,44], with *Lcb. rhamnosus* HN001 in four studies [38,39,40,42], and with *L. acidophilus* and *Bifidobacterium* in two studies [41,43]. In conclusion, no metabolic or inflammatory response improvement was observed [39,43], except in the study by Hassain et al. [41], who observed an improvement in post-GDM women through the modulatory effect on intestinal dysbiosis.

## 9. The Action of Probiotics Regarding the Prevention and Evolution of GDM

During pregnancy, changes in the bacterial composition of the intestinal microbiome lead to a pro-inflammatory status by increasing chemocytokines, an increase in pregnant women’s weight, and an increase in insulin resistance. All of this leads to a “diabetogenic” status or metabolic syndrome-like phenotype that provides a caloric and energetic supplement for the development of the fetus, especially in the third trimester of pregnancy, and stimulates energy storage in adipose tissue [65,113].

This balance is very fragile due to the action of placental insulinase, the increased resistance to insulin, as well as the impossibility of the mother’s body to secrete additional insulin, causing the pregnant woman to develop GDM [114]. Monitoring the *Bacteroidetes/Firmicutes* relationship during pregnancy defines the diabetogenic phenotype, and therapeutic intervention at this level can certainly be crucial. This was observed by the existence of some pro-diabetogenic species (*Prevotella* subsp. and *Bacteroides fragilis* lipopolysaccharide—LPS) and some anti-diabetogenic species (*Bacteroides thetaiotaomicron*) [65].

Alterations of maternal gut microbiota by probiotic intervention during pregnancy in mice directly affect fetal development, placental morphogenesis, and nutrient transport capacity. In addition, *B. breve* modifies the fetal liver transcriptome to restore fetal glycemia, a particularly important mechanism in fetal growth [69].

Following the analysis of 21 RCTs (Table 4), totaling 4036 patients, probiotic intervention with *Lcb. rhamnosus* GG/HN001 was observed in seven studies [44,45,52,54,56,62,64], with *L. acidophilus* in 12 studies (solely or in combination with *Bifidobacterium*) [47,49,50,51,53,55,57,58,59,60,61,63,64], and with *Lgb. salivarius* (formerly *Lactobacillus salivarius*) in two studies [46,48].

Probiotic supplementation among patients with GDM demonstrated a decrease in FPG and serum insulin levels in seven studies [47,49,55,57,58,59,63], some beneficial effects in three studies [51,60,62], and no effects in five studies [50,54,56,61,64]. Regarding the beneficial effect of probiotic intervention on glucose metabolism, it is possible in normal-weight women [55,62] and in overweight/obese women [60]. In contrast, other studies showed no benefits on glucose metabolism in overweight/obese women [39,54,56].

The administration of probiotics to pregnant women with GDM for 6–8 weeks represents a potential metabolic therapy to reduce insulin resistance in those patients diagnosed with GDM [115], while treatment under 4 weeks did not influence maternal FPG or metabolic profile [46]. Probiotic use in women with GDM observed a negative association with serum fasting insulin and HOMA-IR, with no significant correlation with FPG [105]. Several studies observed a decrease in FPG in pregnant women with GDM [47,49,57,58,59,63]. Instead, Zhang et al. [116], in a meta-analysis of 12 randomized trials, observed in pregnant women without GDM that probiotic supplementation significantly reduced the incidence of GDM, FPG, HOMA-IR insulin concentration, and the quantitative insulin sensitivity test index, with no effect on the oral glucose tolerance test (OGTT). The study concluded that probiotics improved glycemic control and the reduced the incidence of GDM.

Studies have shown that consuming probiotics containing multiple strains versus those with single strains decreased insulin, glucose, and HOMA-IR levels [117,118]. One study showed that food probiotics are more effective than those that come from supplements, which have different formulations and forms of presentation [98], while another study noted a decrease in the risk of secondary metabolic syndrome such as GDM, PE, or excess weight after probiotic interventions [119].

## 10. The Roles of Probiotics in the Prevention of Preeclampsia

A possible explanation of the pathogenesis of preeclampsia (PE) is the action of plasma lipopolysaccharides (LPS) derived from the intestine on Toll-like receptors, with a pro-inflammatory role in exacerbating cardiovascular dysfunction. Anti-inflammatory probiotic intervention reduces the plasma concentration of LPS and plasma trimethylamine-N-oxide (TMAO), thus improving endothelial oxidative stress [120]. Identifying microbial communities at the placental, intestinal, or oral levels will be useful in distinguishing potential contamination (especially placental) from normal microbiota related to gestational age and guiding appropriate metabolomic intervention for this pathology.

The role of the placental microbiome in the pathogenesis of PE remains unclear, although the placental presence of bacteria originating from the oral or gut microbiota has been detected [121]. At the end of pregnancy, the intracellular protection mechanism against pathogens at the placental level is carried out by decidual natural killer cells (dNK). The role of dNK cells in the determinism of PE can be achieved by regulating maternal-fetal immune tolerance, vascular remodeling, and the invasive action of the trophoblast [122]. The presence of bacteria such as *Helicobacter pylori* in the placentas of women with PE [123] or *Listeria monocytogenes*, *Toxoplasma gondii*, as well as *Escherichia coli* in the extravillous trophoblast [124], has been contested by a series of analytical studies that deny the placental presence of bacterial DNA sequences [125,126]. Another study showed that, in PE pathogenesis, cysteine, through endothelial dysfunction, exerts an adverse effect on the fetoplacental unit, together with the metabolic dysfunction of vitamin B6; probiotic supplementation may help prevent this condition [100].

Three biomarkers are more important for diagnosing and monitoring PE: PAPP-A, PlGF, and soluble fms-like tyrosine kinase-1 (sFlt1) [127]. As a result, the evaluation of the effectiveness of any probiotic intervention will have to follow the evolution of these three biomarkers. Adequate control of the diet can ensure a balance in the intestinal microbiota and the integrity of the intestinal wall barrier. The therapeutic effectiveness of the various formulations and the timing of treatment initiation regulate the administration of probiotics regarding pregnancy safety. Nordqvist et al. [128] observed the lack of effect regarding the risk of developing PE when administering probiotics preconceptionally or during the first trimester, compared to the administration of probiotic milk, which showed a potentiated intervention effect.

The genera *Blautia* and *Ruminococcus* proved essential in PE patients’ microbiome changes. Bacteria of the *Blautia* genus are gram-positive, and their abundance is associated with an unfavorable metabolic profile, including changes in glucose tolerance and excessive weight gain, which favors the appearance of obesity and the incidence of GDM [129]. The same mechanism of action was observed in *Ruminococcus*, whose abundance in the intestinal tract was associated with GDM, type 2 diabetes, and PE, the pathogenesis of which intervenes through leptin [130]. In a 2021 Cochrane analysis on GDM prevention, Davidson et al. [131] identified an increase in PE rate secondary to probiotic intervention.

Hypertensive pregnant women show at the periodontal level an exacerbation of the microbiota compared to normotensive ones and changes in the concentrations of nitrate–nitrite-reducing bacteria. Increasing plasma nitrite levels by dietary administration of inorganic nitrates led to a decrease in blood pressure, which indicates the opportunity for possible probiotic intervention [132,133]. In PE, Wang et al. [134] showed that there are differences related to trimesters of pregnancy; thus, they did not observe differences in the abundance of *Proteobacteria*, *Bacteroidetes*, *Actinobacteria*, *Firmicutes*, and *Tenericutes* in the second trimester, instead in the third trimester, a decrease in *Firmicutes* and an increase in *Bacteroidetes* and *Proteobacteria* were observed.

Furthermore, comparing omnivorous diets with vegetarian diets, it was observed that nitrate–nitrite homeostasis does not change at the level of the sublingual plaque, and the decrease in the oral synthesis of nitrites was not associated with an increase in blood pressure [133].

## 11. Other Possible Actions of Probiotics to Improve the Perinatal Outcomes

The mother’s immunometabolic system maintains the balance of the oral, intestinal, vaginal, and placental microbiota through a barrier effect against bacteria, parasites, and viruses. Although the barrier effect is not fully elucidated, supplementing with probiotics is indicated to restore the protective response and repair microflora disorders. A few hours after birth, the newborn creates a normal bacterial flora, objectified by the appearance of bacteria in the feces, and the colonization by *Bifidobacterium* takes place in the first four days of life. Thus, the early colonization with certain strains of *Lactobacilli* and *Bifidobacterium* of the intestine determines subsequent protection against the action of various types of diseases [135].

Supplementation with long-chain polyunsaturated fatty acids during pregnancy, compared to supplementation after birth, proved to have a beneficial effect on fetal neurodevelopment, dependent on the gestational period and independent of the dose used [136].

Changing the vaginal microbiota under the action of probiotics causes a decrease in the pro-inflammatory effects secondary to bacterial action, thus decreasing the rate of premature birth [137]. At the level of intestinal microbiota, the anti-obesity effect mediated by long-chain fatty acids is achieved through a process of thermogenesis of adipocyte cells [138]. Unlike oral probiotics, the adjuvant intervention of vaginal probiotics to prophylactic antibiotic therapy in women with preterm premature rupture of membranes (PPROM) improved perinatal outcomes [139]. The PROMO study on supplementation with probiotics (*L. acidophilus*, *B. lactis*, and *B. lactis* NCIMB 30436) in pregnancies at risk of preterm birth estimated a 50% increase in 3-fucosylactose and 3′-sialylactose in human milk oligosaccharides, with secondary actions at the feto-maternal interface [140].

In a meta-analysis by Grev et al. [141], no evidence was identified regarding the benefit of probiotic intervention in women at low risk for preterm birth or in preterm neonates.

Husain et al. [142], in a randomized, double-blind trial, revealed that oral probiotic intervention with *Lcb. rhamnosus* GR-1 and *Lmb. reuteri* RC-14 during pregnancy does not prevent bacterial vaginosis. The absence of the effect can be explained by resistance to colonization, and as a result, the achievement of colonization with probiotic strains requires an immunomodulation of this resistance, a mechanism currently only being studied [143].

In pregnant patients with periodontitis, the beneficial role of nitrates, which can reduce dysbiosis either through the intake of vegetables or, with better results, using a periodontal gel, has been demonstrated [144]. No improvement was seen in mental health outcomes in obese pregnant women [145].

## 12. Discussions

The analysis of the articles evaluated in this systematic review highlighted the heterogeneous nature of the role of probiotic intervention during pregnancy and lactation to reduce the risk of allergic disease and preeclampsia and prevent obesity, overweight, and GDM. At the basis of all these changes are the immunometabolic mechanisms that undergo a series of transformations related to the different trimesters of pregnancy. These changes are influenced by various factors such as methodology, study design, type of probiotic formulation, time of initiation of the administration, duration of the intervention, and identification of high-risk cases.

The transition from the sterile gastrointestinal tract of the fetus at birth to the one colonized with microorganisms from the environment contributes to the strengthening of the infant’s immune system and the adequate status of its general condition with repercussions in adulthood. The modulation of the intestinal microflora through probiotics determines changes in the functioning of the immune system and, respectively, in the production of cytokines, with various responses depending on the specificity of the different strains of bacteria. Increased resistance to infections is correlated with the immune-stimulatory effects mediated by probiotics and with different levels of cytokine expression related to the action of different Gram-negative or Gram-positive strains. The action of probiotics includes the inhibition of the production of pro-inflammatory cytokines, with different responses to pathogens [146].

The proper use of probiotics ensures a balance in terms of the bacterial composition at the level of the gut microbiota, with the ultimate goal of reducing the action of certain diseases on the body. Andrés et al. [147] analyzed infant fecal samples using 16S ribosomal RNA gene sequencing and multiplexing techniques, showing that the genus *Bifidobacterium* was the most frequently identified (>50% of sequences) pre- and post-probiotic intervention. Thus, the anti-inflammatory, immunomodulatory effects of three probiotic strains [*B. longum* subsp. *longum* 35624 (formerly *B. longum* subsp. *infantis* R0033), *L. helveticus* R0052, and *B. bifidum* R0071] were demonstrated by the increase in the *B. lactis* NCIMB 30436 (formerly *B. infantis*) group of the IL-10/IL-12 ratio and the TNF-α/IL-10 ratio in the *L. helveticus* group.

In recent years, a series of *Lactobacilli* and *Bifidobacterium* strains have been isolated with immunomodulatory roles, exerted by increasing innate immunity secondary to the increase in cytotoxicity of natural killer cells (NK) and phagocytosis processes initiated by macrophages, and by increasing adaptive immunity at the level of enterocytes and dendritic cells, Th1, Th2, and Treg, as well as preventing damage to the gastric and intestinal mucosa [148]. Regulation of the intestinal microflora by probiotics contributes to reducing clinical manifestations of lactose intolerance; they have antagonistic effects on the action of pathogens and help the bioavailability of nutrients. Probiotic interventions initiate immune responses, limit the inflammatory process, and modulate the action of NK cells and the response of T-helper cells [149].

The results regarding the intervention of probiotics in reducing the incidence of eczema or atopic dermatitis are controversial; two studies found no measurable effect, and two studies found a decrease in the rate of eczema or atopic dermatitis. However, prenatal administration reduced the rate of allergies in high-risk infants [150]. In a meta-analysis conducted by Garcia-Larsen et al. [151], which included 19 studies with 4076 patients, the beneficial effects of probiotic supplementation were identified, reducing the risk of eczema (risk ratio—0.78).

Probiotic intervention in obese women compared to the placebo group indicated a higher average FPG, an incidence of PE of 9.2% versus 4.9%, excessive weight gain in 32.5% of cases versus 46%, and a small for gestational age (SGA) rate of 2.4% versus 6.5% [56]. However, Lindsay et al. [46] did not observe any influence of a four-week probiotic intervention on maternal FPG or metabolic profile in obese women. Thus, the probiotic component and duration of intervention appear to be parameters that my influence the results of the studies.

In a randomized controlled trial (HUMBA), Okesene-Gafa et al. [152] studied 230 women without diabetes and with a BMI ≥30 kg/m^2^, who were treated with probiotics with formulations containing *Lcb. rhamnosus* GG and *B. lactis* Bb-12, between 12^+0^ and 17^+6^ weeks of gestation. No significant differences were observed in excessive gestational weight gain or neonatal weight compared to the control group.

The evaluation of the effects of probiotic interventions in the assessment of metabolic disorders is based on various markers such as free fatty acids, alanine/aspartate transaminases, plasma cholesterol, and proteomics elements of inflammatory processes and immunometabolic pathways.

A reduction in adiposity was identified in women with GDM [40,42], with a possible effect on the DNA methylation status of obesity promoters and genes related to weight gain (with early intervention in pregnancy at 18 weeks) [37]. In addition, early modulation of the gut microbiota may reduce excessive weight gain in the first years of life [44].

Probiotics containing *L. gasseri* may have beneficial effects in overweight/obese patients by acting on adipose tissue mass and preventing metabolic disorders. Probiotic intervention with *Lcb. rhamnosus* GG and *B. lactis* observed a reduction in abdominal adiposity at 6 months postpartum [153,154]. In late pregnancy, adiposity was reduced in women with GDM after probiotic administration with *Lcb. rhamnosus* HN001 and *B. animalis* subsp. *lactis* 420 [40]. Furthermore, probiotics containing the same formulation, solely and in combination with fish oil, from early pregnancy until 6 months postpartum in overweight/obese women lowered the adiposity [42].

Probiotic interventions reduced the prevalence of GDM in overweight/obese patients [45] or the elderly and patients without GDM [44,52]. However, other studies have demonstrated the absence of any effect in reducing the risk of GDM [54,56,61]. After the probiotic intervention, some studies have shown a reduction in insulin resistance [49,59,63], and others reported no differences in the HOMA-IR index [50,54]. Probiotics may improve glycemic control during the second [60] or third trimester [50].

Supplementation with probiotics from a single genus (e.g., *Lgb. salivarius* UCC118 [48]) has not been shown to have a beneficial effect on glycemic control; instead, the use of preparations containing *S. thermophilus*, *L. acidophilus*, *Lacticaseibacillus paracasei *(formerly *Lactobacillus paracasei*), *Lactiplantibacillus plantarum* (formerly *Lactobacillus plantarum*), *L. helveticus* NCIMB 30440 (formerly *Lactobacillus delbrueckii* subsp. *bulgaricus*), and *Bifidobacterium* [e.g., *B. breve*, *B. longum*, *B. lactis* NCIMB 30436 (formerly *B. infantis*)] at a dose of 1.13 × 10^11^ for 8 weeks was accompanied by a much better response [50].

Studies in the literature are currently contradictory regarding the benefit of the administration of probiotics in pregnant women with GDM. The administration of probiotics does not influence the incidence of gestational diabetes, with the only recorded action being a minimal decrease in FPG by increasing insulin sensitivity [155].

In the first trimester of pregnancy, the relationship between adipokine levels and energy metabolism is influenced by the action of the *Ruminococcaceae* and *Lachnospiraceae* genera. This observation suggests that the intervention of probiotics on the intestinal microbiota can actively intervene in metabolic processes [156]. Regarding the abundance of the genus *Bifidobacterium*, especially in the subgroup of obese pregnant women, some studies identify a protective effect on PE [99,157], while others indicate a harmful effect [158].

There is currently no set of recommendations regarding probiotic supplements, with the answers being different due to the increased variability regarding their use. No adverse effects were reported during probiotic interventions in mothers or infants, demonstrating the safety profile of their use during pregnancy and breastfeeding.

The inclusion in this review of studies that examined probiotic interventions in both pregnant mothers and their infants allowed for a better analysis of their effectiveness. Other strengths of this review are represented by the large number of randomized clinical trials and the many diseases analyzed. The Cochrane risk of bias tool was used to reduce the risk of bias. However, the relatively large number of formulations, the different administration periods, the different randomization methods, and the small number of analyzed databases represent a series of limitations. Although the results are promising regarding the beneficial potential for prevention and immunometabolic modulation, additional clinical evidence is needed to demonstrate efficacy in high-risk patient groups.

## 13. Conclusions

Interventions with probiotics have effects mediated through the maternal microbiome on the maternal-fetal immunometabolic status, ultimately leading to the improvement of perinatal outcomes and a reduction in the risk regarding the severity of certain conditions (gestational diabetes, overweight/obesity, preeclampsia, allergies, eczema). The use of probiotic supplements during pregnancy depends on the dose, formulation, gestational age at which the treatment was initiated, and duration of administration, thus intervening, to some extent, the prophylactic and therapeutic management of various ailments encountered during pregnancy, with subsequent effects on the newborn.

## 14. Take Home Messages

Probiotic intervention during pregnancy and lactation may help reduce the risks of allergic diseases, preeclampsia, obesity, overweight, and GDM. However, study outcomes vary due to differences in methodologies, probiotic types, intervention timing, and risk identification.Infant immune system development. Transitioning from a sterile fetal gut to one colonized by environmental microorganisms is essential for strengthening the infant’s immune system, influencing long-term health.Immune system modulation. Probiotics can modulate the gut microbiota, affecting cytokine production and immune responses. The specific effects depend on the bacterial strains, enhancing infection resistance and reducing pro-inflammatory cytokine production.Probiotics and gut health. Appropriate probiotic use supports a balanced gut microbiota, reducing the impact of certain diseases. Studies indicate that probiotics, particularly *Bifidobacterium*, may exert anti-inflammatory effects and improve the IL-10/IL-12 ratio.Immunomodulatory roles. Various strains of *Lactobacilli* and *Bifidobacterium* have immunomodulatory effects, enhancing both innate and adaptive immunity, alleviating lactose intolerance, and increasing nutrient bioavailability.Mixed evidence on allergic disease reduction. Evidence regarding probiotics’ effect on reducing eczema and atopic dermatitis is mixed, though prenatal use may lower allergy rates in high-risk infants.Conflicting results on metabolic health. Studies show varying results on probiotics’ impact on obesity, FPG, excessive weight gain, and metabolic health in pregnant women. The probiotic strain, intervention duration, and other factors influence these outcomes.Limited effects on gestational diabetes. The benefit of probiotic use in reducing GDM remains unclear. Some studies show minor improvements in insulin sensitivity, while others report no effect.No standard probiotic recommendations. Due to the variability in study findings, there are no standardized recommendations for probiotic supplementation during pregnancy. However, probiotics appear safe for use in pregnant women and infants.

## Figures and Tables

**Figure 1 nutrients-17-01535-f001:**
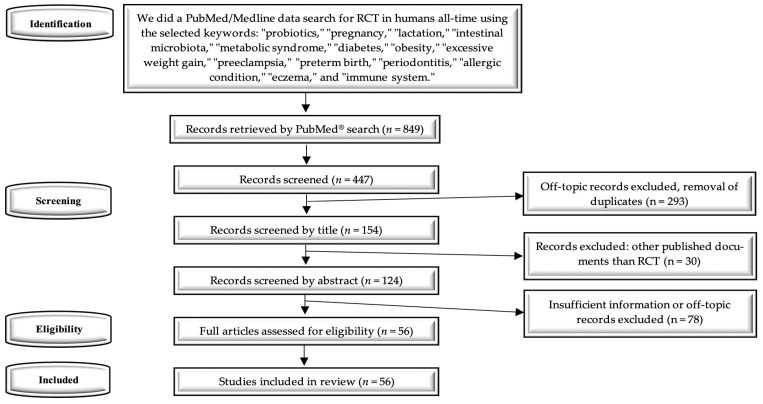
PRISMA diagram—systematic search and study selection process.

**Figure 2 nutrients-17-01535-f002:**
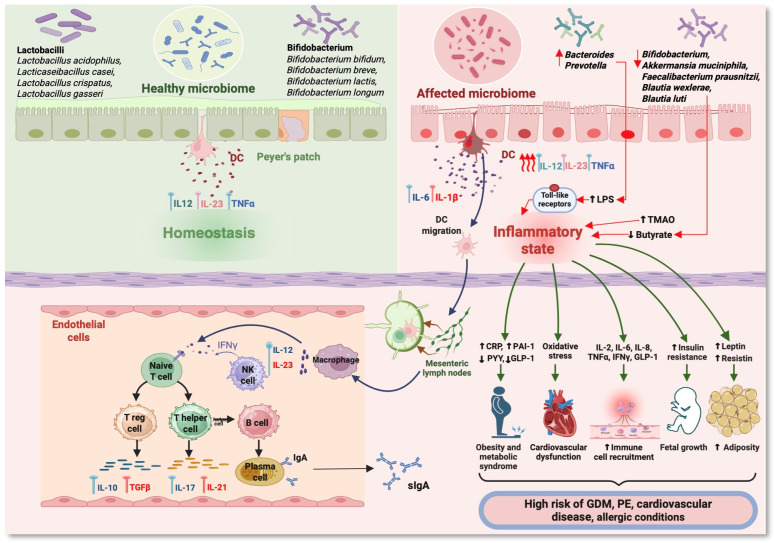
Immunometabolic pathways in pregnancy related to changes in the maternal microbiome. (DC—dendritic cell; LPS—lipopolysaccharides; TMAO—trimethylamine N-oxide; sIgA—secretory immunoglobulin A; CRP—C-reactive protein; PAI-1—plasminogen activator inhibitor-1; PYY—peptide YY; GLP-1—glucagon-like peptide-1; GDM—gestational diabetes mellitus; PE—preeclampsia; IFN*γ*—interferon *γ*; TNF*α*—tumor necrosis factor alpha; IL—interleukin) (Figure created with https://www.BioRender.com, accessed on 1 August 2024).

**Table 1 nutrients-17-01535-t001:** Synopsis of the RCTs on probiotic intervention on the immune response during pregnancy and lactation.

Authors, Reference, Year	Study Design	Cases (Probiotics/Placebo)	BMI	Age	Probiotic Intervention	DoseCFU/Day	Intervention Period	Outcomes	Results	Overall Bias Risk
Rinne [9], 2005	D-B, P-CCT	96	N/A	N/A	- *Lacticaseibacillus rhamnosus* GG	>1 × 10^10^	4 wks before expected delivery, and 6 m postpartum in infants	To assess the impact of intervention and breastfeeding on gut microecology.	Probiotics in the mother’s diet before delivery and in the infant’s diet during breastfeeding may positively influence the maturation process of gut immunity.	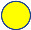
Marschan [10], 2008	D-B, P-CCT	98 (52/46)	N/A	N/A	- *Lcb. rhamnosus* (GG, LC705) - *B. breve* Bb99 - *Propionibacterium freudenreichii* subsp. *shermanii* JS	5 × 10^9^5 × 10^9^2 × 10^8^3 × 10^9^,2 cps/day	4 wks before delivery, and < 6 m postnatally in infants (family history of allergy)	Effect on in vivo cytokine, antibody, and inflammatory responses in allergy-prone infants.	The elevation of IgE, IgA, and IL-10 characterized probiotic-induced low-grade inflammation. ↑ Plasma CRP levels at 6 m was associated with a decreased risk of eczema and allergic disease at 2 yrs.	
Kopp [11], 2008	Prospective, D-B, P-CCT	68 (40/28)	N/A	N/A	- *Lcb. rhamnosus* GG	5 × 10^9^, twice daily	4–6 wks before expected delivery, and postnatal for 6 m	Proliferative response and cytokine release in cultures of isolated mononuclear cells.	No difference in proliferative capacity or cytokine pattern of maternal or neonatal cord blood cells in response to IL-2, β-lactoglobulin, or LGG.	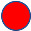
Rautava [12], 2012	D-B, P-CCT	43 (28/15)	N/A	N/A	- *Lcb. rhamnosus* GG - *B. lactis*	1 × 10^9^, each	14 days before elective cesarean at full-term	Innate immune gene expression profiles in the placenta and fetal gut may be modulated during late pregnancy.	Significantly modulated the expression of TLR-related genes both in the placenta and in the fetal gut.	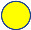
Rø [13], 2017	D-B, P-CCT	415 (211/204)	29.3/29.7	30.8/30.4	- *Lcb. rhamnosus* GG - *B. animalis* subsp. *lactis* Bb-12 - *L. acidophilus* La-5	5 × 10^10^5 × 10^10^5 × 10^9^	From 36 wks to 3 m postnatally while breastfeeding	Modifying Th cell proportions could mediate the preventive effect of probiotics on AD.	Perinatal intervention with probiotics reduced the proportion of Th22 cells in 3-month-old children, which may explain the partial protective effect on Alzheimer’s disease.	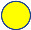
Forsberg [14], 2020	Prospective, D-B, P-CCT	88	N/A	N/A	- *Limosilactobacillus reuteri* (formerly *Lactobacillus reuteri*)	1 × 10^9^, plus ω-3 PUFA 3 capsules twice daily	From 20 wks until delivery	Maternal peripheral immunity.	Some immunomodulatory effects were observed among circulating activated and resting T cells.	
Soukka [15], 2023	D-B, P-CCT	439fish oil + probiotics n = 109 placebo n = 110	28.8/28.9	30.7/31	- *Lcb. rhamnosus *HN001 - *B. animalis *subsp. *lactis* 420	1 × 10^10^,each	From early pregnancy (<18 wks) until 6 m postnatally in women with pre-pregnancy BMI ≥ 25 and ≥30 kg/m^2^	Modified concentrations of colostrum immune mediators and associations with perinatal clinical factors on mothers with overweight/obesity.	The fish oil + probiotics group had higher levels of IL-12p70, FLT-3L, and IFNα2. Intervention exerted a minor effect on concentrations of colostrum immune mediators and may contribute to immune system development in the infant.	
Killeen [16], 2024	D-B, P-CCT	72 (36/36)	36/36	36/36	- *B. lactis* NCIMB 30435 (formerly *B. longum* subsp. *longum* 1714)	>1 × 10^9^	From 16–20 wks until delivery	Change in IL-10 production after stimulation with lipopolysaccharide.	Not alter cytokine production by maternal PBMCs in response to PAMPs or anti-CD3/28/2	

PBMC—peripheral blood mononuclear cells; PAMP—pathogen-associated molecular patterns; FLT-3L—FMS-like tyrosine kinase 3 ligand; AD—atopic dermatitis; TLR—Toll-like receptor; D-B—double-blind; P-CCT—placebo-controlled clinical trial; ↑—increase; *Lcb.*—*Lacticaseibacillus*; *B.*—*Bifidobacterium*. Overall bias risk: 

 Low risk; 
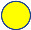
 Moderate risk; 
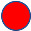
 High risk.

**Table 2 nutrients-17-01535-t002:** Synopsis of the RCTs on probiotic intervention on allergic conditions and eczema.

Authors, Reference, Year	Study Design	Cases (Probiotics/Placebo)	BMI	Age	Probiotic Intervention	Dose CFU/Day	Intervention Period	Outcomes	Results	Overall Bias Risk
Kalliomäki [17], 2001	D-B, P-CCT	159 (270/270)	N/A	N/A	- *Lcb. rhamnosus* GG	>1 × 10^9^	For 2–4 wks before expected delivery, and 6 m postnatally in women if breastfeeding	Preventing early atopic disease.	Effective in prevention of early atopic disease in children at high risk.	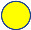
Kukkonen [18], 2007	D-B, P-CCT	1223 (610/613)	N/A	N/A	- *Lcb. rhamnosus* (GG, LC705) - *B. breve* Bb99 - *P. freudenreichii* subsp. *shermanii* JS	5 × 10^9^5 × 10^9^2 × 10^8^2 × 10^9^	For 2–4 wks before delivery and 3 m postnatally in infants	Preventing allergic diseases.	No effect on all allergic diseases by age 2 yrs, but prevented eczema and AD; results suggest an inverse association between atopic diseases and gut colonization by probiotics.	
Abrahamsson [19], 2007	Prospective, multicenter, D-B, P-CCT	232 (117/115)	N/A	N/A	- *Limosilactobacillus reuteri*	1 × 10^8^	From 36 wks until delivery, and postnatal for 12 m to their infants	Allergic disease with or without a positive skin prick test or circulating IgE to food allergens.	No preventive effect of eczema; the treated infants had less IgE-associated eczema at 2 yrs and possibly a reduced risk of developing later respiratory allergic disease.	
Huurre [20], 2008	D-B, P-CCT	171 (72/99)	N/A	N/A	- *Lcb. rhamnosus GG* (ATCC 53103) - *B. lactis* Bb-12	1 × 10^10^, each	From the first trimester to the end of exclusive breastfeeding	Maternal allergy prevention, infant sensitization.	Infants of atopic mothers, when breastfed exclusively over 2.5 m or a total over 6 m, had a higher risk of AS at 12 m; the risk was reduced by the use of probiotics.	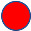
Niers [21], 2009	D-B, P-CCT	156 (78/78)	N/A	31.4/32.3	- *B. bifidum* W23 - *B. lactis* W52- *Lactococcus lactis* W58	1 × 10^9^, each	During the last 6 wks before expected delivery, and postnatal for 12 m to their infants	Effects on the development of eczema in the first 2 yrs, and on early microbial colonization and immune responses.	Preventive effect of early administration on the incidence of eczema in the first 3 m, with significant changes in the intestinal microbiota and decreased IL-5 production.	
Kuitunen [22], 2009	D-B, P-CCT	1223 (991 children) (445/446)	N/A	N/A	- *Lcb. rhamnosus* (LC705, GG)- *B. breve* Bb99, - *P. freudenreichii* subsp. *shermanii* JS	5 × 10^9^5 × 10^9^2 × 10^8^2 × 10^9^2 cps/day	During last month, and 6 m postnatally in infants	Preventing allergic diseases.	No allergy-preventive effect during the last month of pregnancy and for infants 6 m after birth, with evaluation at 5 yrs. Protection only to cesarean-delivered children.	
Kim [23], 2010	D-B, P-CCT	112 (57/55)	N/A	29.9/29.5	- *B. bifidum* BGN4, - *B. lactis* AD011 - *L. acidophilus* AD031	1.6 × 10^9^, each	For 4–8 wks before delivery, and 6 m postnatally in women if breastfeeding	Preventing the development of eczema and AS against food allergens in infants at high risk of atopic disease.	Beneficial effect to prevent development of eczema in infants at high risk during their first year of life.	
Dotterud [24], 2010	D-B	415 (138/140)	N/A	29.9/29.7	- *Lcb. rhamnosus* GG - *L. acidophilus* La-5 - *B. animalis* subsp. *lactis* Bb-12	5 × 10^10^5 × 10^9^5 × 10^10^	From 36 wks until delivery to 3 m postnatally during breastfeeding	Preventing AS or allergic diseases during the child’s first 2 yrs.	Reduced the cumulative incidence of atopic dermatitis but did not affect AS.	
Boyle[25], 2011	Controlled trial	250 (125/125)	N/A	N/A	- *Lcb. rhamnosus* GG	1.8 × 10^10^	From 36 wks until delivery	Preventing the development of eczema.	No reduced risk of eczema or IgE-associated eczema. No change in cord blood immune markers. ↓ Breast milk CD14/IgA levels.	
Wickens [26], 2012	D-B, P-CCT	474 (315/159)	N/A	N/A	- *Lcb. rhamnosus* HN001 - *B. animalis* subsp. *lactis* HN019	6 × 10^9^9 × 10^10^	From 35 wks until delivery, and 6 m postnatally in women if breastfeeding, and until 2 yrs in infants	Prevalence of eczema and allergic diseases at 4 yrs.	Reduced the cumulative prevalence of eczema and rhinoconjunctivitis at 4 years.	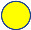
Ismail [27], 2012	D-B, P-CCT	98 (50/48)	N/A	N/A	- *Lcb. rhamnosus* GG	1.8 × 10^10^	From 36 wks until delivery	Preventing the development of eczema in infants at high risk of allergic disease.	Administration of probiotics to mothers failed to modulate the diversity of infant gut microbiota in the first week of life.	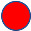
Kuitunen [28], 2012	Prospective, D-B, P-CCT	346 (166/180)	N/A	N/A	- *Lcb. rhamnosus* (GG, LC705) - *B. breve* Bb99 - *P. freudenreichii* subsp. *shermanii* JS	5 × 10^9^5 × 10^9^2 × 10^8^2 × 10^9^	From 36 wks until delivery in women with allergic disease, and postnatal for 6 m to their infants	Preventing allergic disease/IgE-associated allergic disease cumulatively and eczema/AD until ages 2 and 5.	Increased IL-10 and decreased casein IgA antibodies in colostrum. Minor effects on allergy development in children until the ages of 2 and 5.	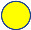
Rautava [29], 2012	Prospective, parallel group, D-B, P-CCT	241 (163/78)	N/A	N/A	(*1*)—- *Lcb. rhamnosus* LPR - *B. longum* BL999 (LPR + BL999) (*2*)—- *Lcb. paracasei* ST11 (formerly *Lactobacillus paracasei*)- *B. longum* BL999 (ST11 + BL999)	1 × 10^9^, each	From 2 m until delivery, and 2 m postnatally if breastfeeding in women with allergic disease/AS.	Cumulative incidence of eczema in the infant up to 2 yrs. AS in the infants.	Prevention probiotics administered prenatally/postnatally are safe and effective in reducing the risk of eczema and have no effect on the risk of AS in infants with allergic mothers.	
Allen [30], 2014	D-B, P-CCT, parallel-group	454 (220/234)	N/A	N/A	- *Ligilactobacillus salivarius* CUL61 - *Lcb. paracasei* CUL08 - *B. animalis* subsp. *lactis* CUL34, - *B. bifidum* CUL20	6.25 × 10^9^1.25 × 10^9^1.25 × 10^9^1.25 × 10^9^	From 36 wks until expected delivery, and 6 m postnatally in infants	Cumulative frequency of diagnosed eczema at 2 yr follow-up.	Prevent AS to common food allergens and reduce the incidence of atopic eczema in early childhood.	
Kim [31], 2015	D-B, P-CCT	123 (60/63)	N/A	N/A	- *B. bifidum* W23, - *B. animalis* subsp. *lactis* W52- *Lactococcus lactis* W58	1 × 10^9^, each	For 6 wks until delivery (family history of allergic disease), and postnatal for 12 m to their infants	Preventing the development of eczema until 2 yrs.	Induced higher levels of lactate and SCFAs and lower levels of lactose and succinate. Bacterial metabolites may play a role in developing the immune system and have temporary preventive effects on the development of eczema.	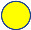
Simpson [32], 2015	D-B, P-CCT	415 (211/204)	N/A	N/A	- *Lcb. rhamnosus* GG - *L. acidophilus* La-5 - *B. animalis* subsp. *lactis* Bb-12	5 × 10^10^5 × 10^9^5 × 10^10^	From 36 wks before expected delivery until 3 m postnatally	Incidence of AD, allergic rhinoconjunctivitis, AS; prevalence of asthma at 12 m.	Long-term reduction in the cumulative incidence of AD, but not other allergy-related conditions (asthma, AS).	
Barthow [33], 2016	D-B, P-CCT	400 (200/200)	31.9/31.6	31.3/31.7	- *Lcb. rhamnosus* HN001	6 × 10^9^	From 14–16 wks until delivery, and 6 m postnatally if women breastfeeding	Prevalence of infant eczema and AS, GDM, maternal postpartum depression and anxiety.	Probiotic action prevents infant eczema, atopic disease, and GDM at one year.	
Wickens [34], 2018	D-B, P-CCT, parallel-group	423 (212/211)	N/A	N/A	- *Lcb. rhamnosus* HN001	6 × 10^9^	From 14–16 wks before delivery, and 6 m postnatallyif women breastfeeding	Prevalence of infant eczema and AS.	Maternal and infant probiotic supplementation may be effective for preventing infant eczema. No significant effect on eczema, wheeze, or AS in the child by age 12 m. No effect on the levels of breast milk proteins, TGFβ1, TGFβ2, and IgA.	
Davies [35], 2018	D-B, P-CCT, parallel-group	452 (220/234)	N/A	N/A	- *Lgb. salivarius* CUL61 - *Lcb. paracasei* CUL08 - *B. animalis* subsp. *lactis* CUL34 - *B. bifidum* CUL20	1 × 10^10^, each	From 36 wks until delivery, and postnatal for 6 m to their infants	Prevalence of eczema and asthma at or before 2 yrs of age.	The 5-year electronic follow-up did not find support for the effect of early use of probiotics on childhood eczema or asthma.	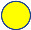
Shipton [36], 2024	D-B, P-CCT	411 (207/204)	32.6/32.9	37/37.9	- *Lcb. rhamnosus* GG - *B. animalis* subsp. *lactis* Bb-12	NS	From <16 wks until delivery, in women with pre-pregnancy BMI of >25 kg/m^2^	Prevalence of childhood allergic diseases.	Infants do not appear to have any pediatric health advantages regarding allergic diseases.	

SCFA—short-chain fatty acids; SGA—small for gestational age; wks—weeks; m—months; yrs—years; ↓—decrease; S—atopic sensitization; AD—atopic dermatitis; *B.*—*Bifidobacterium*; *P.*—*Propionibacterium*; *Lcb.*—*Lacticaseibacillus*; *Lgb.*—*Ligilactobacillus*; *L.*—*Lactobacillus*. Overall bias risk: 

 Low risk; 
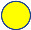
 Moderate risk; 
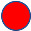
 High risk.

**Table 3 nutrients-17-01535-t003:** Synopsis of the RCTs on probiotic intervention on obesity and excessive gestational weight gain.

Authors, Year	Study Design	Cases (Probiotics/Placebo)	BMI	Age	Probiotic Intervention	Dose CFU/Day	Intervention Period	Outcomes	Results	Overall Bias Risk
Vähämiko [37], 2019	D-B, P-CCT	15 (7/8)	21.7/24.5	29.5/28.6	- *Lcb. rhamnosus* GG - *B. lactis* Bb-12	1 × 10^9^, each	From early pregnancy (≤18 wks) and family history of allergic disease	DNA methylation status of the promoters of obesity and weight gain-related genes in mothers and their children.	Probiotics may affect the DNA methylation status of certain promoters of obesity and weight gain-related genes.	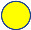
Houttu [38], 2020	D-B, P-CCT	439probiotics + placebo n = 110	N/A	30.6	- *Lcb. rhamnosus* HN001 - *B. animalis* subsp. *lactis* 420	1 × 10^10^, each; plus 2 fish oil capsules	From early pregnancy (≤18 wks) until 6 m postnatally in women with pre-pregnancy BMI of >25 kg/m^2^	Impact on serum/vaginal inflammatory and metabolic proteins and relation to the onset of GDM.	The intervention did not impact the proteins, but obesity and GDM may modify the effect.	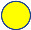
Mokkala [39], 2021	D-B, P-CCT	358probiotics +placebo n = 91	28	31	- *Lcb. rhamnosus* HN001 - *B. animalis* subsp. *lactis* 420	1 × 10^10^, each; plus 2 fish oil capsules	From the first visit (≤18 wks) until 6 m postnatally in women with pre-pregnancy BMI of >25 kg/m^2^	Metabolic alterations in pregnant women with overweight or obesity.	In women with GDM, no effects on any metabolites. Fish oil and probiotics modified serum lipids (VLDL, LDL).	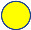
Pellonperä [40], 2021	P-CCT intervention pilot	439 (211/204)	N/A	N/A	- *Lcb. rhamnosus* HN001 - *B. animalis* subsp. *lactis* 420	1 × 10^10^, each; plus 2 fish oil capsules	In late pregnancy >35 wks	Gestational weight gain and body composition.	Adiposity was reduced in women with GDM, irrespective of the dietary intervention.	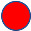
Hassain [41], 2022	12-wks, D-B, P-CCT, parallel-group	132 (66/66)	N/A	N/A	- *L. acidophilus* - *L. lactis* - *Lacticaseibacillus casei* subsp. BCMC 12313 - *B. bifidum* - *B. lactis* NCIMB 30436 (formerly *B. infantis*) - *B. lactis* NCIMB 30435	3 × 10^10^, twice daily	12 wks (at 4–8 wks postnatallyin asymptomatic post-GDM women)	Metabolic and inflammatory effects.	Improved metabolic and inflammatory outcomes in post-GDM women by modulating gut dysbiosis.	
Saros [42], 2023	D-B, P-CCT	439 (211/204)	N/A	N/A	- *Lcb. rhamnosus* HN001 - *B. animalis* subsp. *lactis* 420	1 × 10^10^, each; plus 2 fish oil capsules	From early pregnancy (≤18 wks) until 6 m postpartum	The tendency of children <24 m to become overweight; change body fat percentage.	Probiotics, alone or in combination with fish oil, during pregnancy to overweight/obese women lowered overweight incidence.	
Halkjær [43], 2023	D-B	50	N/A	N/A	Vivomixx—*L. acidophilus, Lcb. paracasei, Lactiplantibacillus plantarum, L. helveticus* NCIMB 30440, *B. lactis* NCIMB 30435, *B. breve, B. lactis* NCIMB 30436, *S. thermophilus*	4.5 × 10^11^	From 14–20 wks until delivery in women with pre-pregnancy BMI of >30 kg/m^2^ and <35 kg/m^2^	The effect during pregnancy on the offspring’s metabolic and inflammatory markers and body composition.	No significant effect of supplementation in mothers or babies on metabolic or inflammatory biomarkers.	

*Lcb.*—*Lacticaseibacillus*, *B.*—*Bifidobacterium*, *Lgb.*—*Ligilactobacillus*; *Lmb.*—*Limosilactobacillus*; *L.*—*Lactobacillus*; *P.*—*Propionibacterium*; *S.*—*Streptococcus*. Overall bias risk: 

 Low risk; 
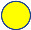
 Moderate risk; 
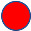
 High risk.

**Table 4 nutrients-17-01535-t004:** Synopsis of the RCTs on probiotic intervention on glucose metabolism and GDM prevention.

**Authors, Year**	**Study Design**	**Cases (Probiotics/Placebo)**	**BMI**	**Age**	**Probiotic Intervention**	**Dose CFU/Day**	**Intervention** **Period**	**Outcomes**	**Results**	**Overall Bias Risk**
Luoto [44], 2010	D-B, P-CCT	159 (77/82)	N/A	N/A	- *Lcb. rhamnosus* GG	>1 × 10^10^	4 wks before expected delivery and postpartum for 6 m	Prevalence of GDM; the impact of perinatal intervention on childhood growth patterns; the development of overweight during a 10-yr follow-up.	↓ Risk of GDM from 34% to 13% moderated the initial phase of excessive weight gain, with impact at 4 yrs. Early gut microbiota modulation may restrain excessive weight gain during the first years of life.	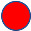
Nitert [45], 2013	Prospective, multicenter, D-B, controlled trial	540 (270/270)	N/A	N/A	- *Lcb. rhamnosus* GG - *B. animalis* subsp. *lactis* Bb-12	>1 × 10^9^, each	From 16 wks until delivery, in women with BMI of >25 kg/m^2^	Prevalence of GDM, weight gain, gut microbiome, macrosomia, infant body composition.	The efficacy of probiotic ingestion from early pregnancy to prevent GDM in overweight/obese women.	
Lindsay [46], 2014	D-B, P-CCT	138/63 (211/204)	32.9/34.1	31.4/31	- *Lgb. salivarius* UCC118	1 × 10^9^	4 wks (at 24–28 wks in women with BMI 30–39.9 kg/m^2^)	Maternal FPG in obese pregnant women.	4 wks of treatment did not influence maternal FPG, the metabolic profile, or pregnancy outcomes.	
Dolatkhah [47], 2015	D-B, P-CCT	64 (29/27)	31.4/29.8	28.1/26.4	- *L. acidophilus* LA-5 - *L. helveticus* NCIMB 30440 - *B. lactis* Bb-12 - *Streptococcus thermophilus* STY-31	>4 × 10^9^	8 wks (at 24−28 wks + 6 days, in women with GDM)	Glucose metabolism, weight gain among GDM patients.	↓ FPG	
Lindsay [48], 2015	D-B, P-CCT	100 (48/52)	N/A	N/A	- *Lgb. salivarius* UCC118	1 × 10^9^	From diagnosis of GDM at <34 wks until delivery	FPG in GDM patients.	No impact on glycemic control or pregnancy outcome.	
Karamali [49], 2016	Prospective, D-B, P-CCT	60 (30/30)	N/A	N/A	- *L. acidophilus * - *Lbs. casei* - *B. bifidum*	2 × 10^9^, each	6 wks (at 24–28 wks, in women with GDM)	Glycemic control, lipid profiles in primigravida with GDM.	↓ FPG, serum insulin, insulin resistance, triglycerides, and VLDL cholesterol.	
Jafarnejad [50], 2016	Multicenter, prospective, D-B, P-CCT	82 (41/41)	N/A	N/A	*VSL#3—S. thermophilus*, *L. acidophilus*, *Lpb. plantarum*, *Lcb. paracasei*, *L. helveticus* NCIMB 30440, *B. breve*, *B. lactis* NCIMB 30435, *B. lactis* NCIMB 30436	1.13 ×10^11^, twice daily	8 wks (from the third trimester until delivery in women with GDM)	Glycemic control and inflammatory parameters.	There were no changes in FPG, Il-10, HbA1c, HOMA-IR, or insulin levels. ↓ IL-6, TNF-∝, and hs-CRP. Modulated inflammatory markers; benefits on glycemic control.	
Halkjaer [51], 2016	D-B, P-CCT	50 (25/25)	22.7/22.0	32.5/30.7	Vivomixx—*L. acidophilus, Lpb. plantarum, Lcb. paracasei, L. helveticus* NCIMB 30440, *S. thermophilus, B. breve, B. lactis* NCIMB 30435, *B. lactis* NCIMB 30436	1.12 × 10^11^/capsule, 2 capsules twice daily	18–24 wks(from 14–20 wks until delivery in women with BMI 30–35 kg/m^2^)	Maternal weight gain, glycated hemoglobin (HbA1c) levels, glucose homeostasis, pregnancy outcomes.	Control weight gain and reduce complications during pregnancy in obese patients.	
Wickens [52], 2017	Two-centre, D-B, P-CCT, parallel	423 (212/211)	25/26	34	- *Lcb. rhamnosus* HN001	6 × 10^9^	> 22 wks(from 14–16 wks, until 6 m postnatally if women still breastfeeding)	Prevalence of GDM.	↓ GDM prevalence, particularly among older patients and those with previous GDM.	
Hajifaraji [53], 2018	D-B, P-CCT	64	N/A	N/A	- *L. acidophilus* LA-5 - *L. helveticus* NCIMB 30440 - *B.* Bb-12 - *S. thermophilus* STY-31	> 4 × 10^9^	8 wks (at 24–28 wk + 6 days, in women with GDM)	Inflammation and oxidative stress biomarkers in newly diagnosed GDM women.	It appears to improve several inflammation and oxidative stress biomarkers in women with GDM.	
Pellonperä [54], 2019	D-B, P-CCT	439 (27/22)	29.3/30.0	30.8/30.4	- *Lcb. rhamnosus* HN001 - *B. animalis* subsp. *lactis* 420	1 × 10^10^	From the first obstetrical visit (<18 wks mean 13.9 ± 2.1 wks) until 6 m postnatally	The incidence of GDM and change in FPG.	No change in FPG, insulin resistance HOMA2-IR index, neonatal birth weight, or maternal weight gain. No benefits in reducing the risk of GDM or improving glucose metabolism in overweight/obese women.	
Sahhaf Ebrahimi [55], 2019	D-B, P-CCT	84 (42/42)	30.7	31.6	- *L. acidophilus* - *B. lactis*	300 mg/probiotic yoghurt (1 × 10^6^)	8 wks	Glycemic control, neonatal outcomes in women with GDM.	↓ FPG, HbA1c, and incidence of macrosomia	
Callaway [56], 2019	D-B, controlled trial	411 (207/204)	31.6/31.9	31.3/31.7	- *Lcb. rhamnosus* GG - *B. animalis* subsp. *lactis* Bb-12	1 × 10^9^, each	From <20 wksuntil delivery, in women with BMI > 25 kg/m^2^	GDM prevention—in overweight/obese pregnant women in T2.	No benefit in GDM prevention. ↓ Excessive weight gain and SGA; no differences in other secondary outcomes.	
Kijmanawat [57], 2019	D-B, P-CCT	57 (28/29)	N/A	N/A	- *L. acidophilus* - *B. bifidum*	1 × 10^9^, each	4 wks (24–28 wks in diet controlled GDM women)	Glycemic control in women with GDM.	↓ FPG and serum insulin levels. ↑ Insulin sensitivity.	
Jamilian [58], 2019	D-B, P-CCT	87 (29/28,Vit. D + probiotic n = 30)	27/28.2	31.2/29.9	- *L. acidophilus* - *Lmb. reuteri*- *Limosilactobacillus fermentum* (formerly *Lactobacillus fermentum*) - *B. bifidum*	2 × 10^9^, each plus Vit. D 50,000 IU/every 2 wks	4 wks (at 24–28 wks, in women with GDM, aged 18–40 yrs)	Metabolic status and pregnancy outcomes in women with GDM.	↓ FPG, serum insulin, triglycerides, HDL/total cholesterol ratio, hs-CRP, malondialdehyde. ↑ Insulin sensitivity check index, HDL-cholesterol, TAC.	
Babadi [59], 2019	D-B, P-CCT	48 (24/24)	26.9/27.3	28.8/29.0	- *L. acidophilus * - *Lbs.* *casei* - *Lmb. fermentum* - *B. bifidum*	2 × 10^9^, each	6 wks (at 24–28 wks in women with GDM)	Gene expression related to insulin and inflammation, glycemic control, lipid profiles, inflammatory markers, and oxidative stress.	Upregulated PPAR-γ, TGF-β, VEGF; downregulated gene expression of TNF-∝. ↓ FPG, serum insulin levels, insulin resistance, VLDL- cholesterol, triglycerides, total/HDL-cholesterol ratio, malondialdehyde. ↑ Insulin sensitivity, HDL-cholesterol, nitric oxide, TAC.	
Asgharian [60], 2020	D-B, P-CCT	65 (211/204)	40/42	29.5/29.4	- *S. thermophilus*- *L. helveticus* NCIMB 30440 - *L. acidophilus* La5 - *B. lactis* Bb-12	1 × 10^7^, 1 × 10^7^5 × 10^8^5 × 10^8^	From 22–24 wks until delivery. At 24 wks extra probiotics (*L. acidophilus* La5, *B. lactis* Bb12)	Maternal plasma glucose, maternal and infant complications in overweight/obese women with no GDM.	Probiotic supplementation has some beneficial effects on glucose metabolism of overweight/obese pregnant women.	
Shahriari [61], 2021	Single-center, D-B, P-CCT	542 (271/271)	30.2/30.2	31.8/32.2	- *L. acidophilus* LA1 - *B. longum* sp54 - *B. bifidum* sp9	>7.5 × 10^9^>1.5 × 10^9^>6 × 10^9^	>26 wks (<12 wks, singleton)	Risk of GDM and maternal/neonatal outcomes	Probiotic supplementation from the first half of the T2 does not reduce the risk of GDM.	
Chen [62], 2021	D-B, P-CCT	348 (172/176)	25.1/25.8	33.1/33.8	- *Lcb. rhamnosus* HN001	6 × 10^9^	From 14–16 wks until delivery	Effects on fasting lipids, insulin resistance, and bile acids (BA).	↓ Conjugated BA, which has a possible role in improving glucose metabolism; has no significant effect on fasting lipids.	
Amirani [63], 2022	D-B, P-CCT	60 (30/30)	N/A	N/A	- *L. acidophilus* - *B. bifidum* - *B. lactis* - *B. lactis* NCIMB 30435 plus selenium	2 × 10^9^, each plus 200 μg/day selenium	6 wks in women diagnosed with GDM	Effects on glycemic status, insulin metabolism, lipid profiles, PPAR-γ, and LDL receptor expression.	↓ FPG, insulin levels, insulin resistance, triglycerides, total/LDL-cholesterol. ↑ Insulin sensitivity, gene expression of PPAR-γ/LDLR.	
Nachum [64], 2024	Multicenter, prospective, D-B, P-CCT	85 (41/44)	N/A	N/A	- *Lcb. rhamnosus*- *L. acidophilus* - *Lcb. paracasei* - *B. bifidum* - *B. lactis* - *S. thermophilus*	>6 × 10^9^/capsule, 2 capsules/day	From the time of diagnosis of GDM until delivery	Maternal glycemic parameters and pregnancy outcomes.	Did not affect the glycemic control of women with GDM.	

T2—second trimester; ↑—increase; ↓—decrease; PPAR-γ—peroxisome proliferator-activated receptor gamma; TGF-β—transforming growth factor beta; VEGF—vascular endothelial growth factor; TNF-∝—tumor necrosis factor alpha; FPG—fasting plasma glucose; TAC—total antioxidant capacity; LDL—low-density lipoprotein; SGA—small for gestational age; BA—bile acids; *Lcb.*—*Lacticaseibacillus*, *B.*—*Bifidobacterium*, *Lgb.*—*Ligilactobacillus*; *Lmb.*—*Limosilactobacillus*; *Lpb.*—*Lactiplantibacillus*; *L.*—*Lactobacillus*; *P.*—*Propionibacterium*; *S.*—*Streptococcus*. Overall bias risk: 

 Low risk; 
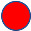
 High risk.

## Data Availability

Data are contained within the article.

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
