# Peer review of "The Influences of Oral Probiotics on the Immunometabolic Response During Pregnancy and Lactation: A Systematic Review"

_nutrients, 2025, doi:10.3390/nu17091535_

Round 1
Reviewer 1 Report
Comments and Suggestions for Authors
A very interesting and innovative study. The authors present a critical and updated perspective on the influence of oral probiotics on the immune response during pregnancy and breastfeeding. The manuscript is well-written, and the methodology used is appropriate. However, I have some comments.
I. Comments:
1. Improve the wording of the study objective.
2. Regarding the observed effects, could maternal obesity play a role? Considering that in many countries (especially in the West), obesity is highly prevalent during pregnancy and breastfeeding.
3. Briefly discuss the role of diet in the effects of probiotics on the inflammatory response.
Author Response
Reviewer 1
A very interesting and innovative study. The authors present a critical and updated perspective on the influence of oral probiotics on the immune response during pregnancy and breastfeeding. The manuscript is well-written, and the methodology used is appropriate. However, I have some comments.
- Comments:
1. Improve the wording of the study objective.
Answer: Thank you for your mention. We added the study objective to the abstract. (Please see the attached manuscript.) Lines 22-26
- Regarding the observed effects, could maternal obesity play a role? Considering that in many countries (especially in the West), obesity is highly prevalent during pregnancy and breastfeeding.
Answer: Thank you for your comment. We have explained how maternal obesity may influence maternal and infant outcomes and the possible mode of probiotic intervention.” (please see the attached manuscript). Section 8
- Briefly discuss the role of diet in the effects of probiotics on the inflammatory response.
Answer: Thank you for your recommendation. We inserted some phrases on the effects of diet on the inflammatory response.” (please see the attached manuscript). Lines 205-222
Kind regards
Reviewer 2 Report
Comments and Suggestions for Authors
nutrients-3578325-peer-review-v1
This is an interesting review paper, where authors have summarized the information on the relation between oral probiotics on the immunometabolic reponse during pregnancy and lactation. In my opinion paper deserve attention form the Editor of the journal and paper can be suggested for publication, however, revision, update and upgrade needs to be considered as option for the improvement of the manuscript by the authors.
In my opinion abstract can be presented better where principal points later discussed in the paper can be summarized and presented.
Ln46: Use of the word Lactobacillus is a quite delicate topic, since in 2020 the classification of former genus Lactobacillus was done. Well, it was suggested in same paper published in 2020 that to avoid misunderstanding between former (till 2020) genus Lactobacillus and new genus Lactobacillus (from 2020, as part of the reclassification), when referencing to works before 2020, English word "lactobacilli" can be applied. Moreover, in 2023, standardized abbreviations for the new genera derivate form former genus Lactobacillus were suggested. doi: 10.1099/ijsem.0.004107; doi: 10.1163/18762891-20230114
Ln65: maybe it will be better to add reference number [5] after Mor et al. in this position. Similar adjustments needs to be made in other parts of the manuscript.
Ln 75: Please, see my previous comment regarding use of term Lactobacillus / lactobacilli.
Ln76: Citing genus Streptococcus and even Propionibacterium is quite delicate, since several species from that 2 genera are well known pathogens. Maybe some additional text needs to be added in this context and clarify the topic.
In Table 1, Please, respect new names for lactobacilli and abbreviate them according to recommendations.
Freudenreichii is one word; ssp. need to be subsp. then shermanii is not with capital S. Strain identification (GG, JS, etc.), do not need to be in italics.
Please, check table 1 and 2 and pay attention on abbreviation, use of ssp. that need to be changed to subsp.; use of coma and full stop, use of capital and not capital, etc. Simply review both tables.
Please, try to provide stain identification for all mentioned species in Table 1 and 2. This is important, since probiotic properties are strain specific and not species specific.
Ln151-153: Since you have mentioned what is predominate in gut microbiota in pregnant women, then will be appropriate to mention how this is different to non-pregnant women. Are there some specific differences? This needs to be pointed out and appropriate references provided. Some points were clarified in the following 6 lines, but topic deserve a bit more attention.
Ln160: Please, correct to Koren et al. [67].
In figure 2, please correct Lactobacillus to lactobacilli, and then abbreviate the mentioned species according to the recommendations.
Please, for the entire manuscript, after formally introducing specific microbial name, written in full, in following occasion, use abbreviation form. However, always when was mentioned for the first time, name needs to be written in full.
In the legend of Figure 2, all abbreviations used in the figure needs to be explained.
Ln170: Zhou et al. [69].
Text into section Ln168-176 can be extended a bit more.
Ln183: Klebsiella is not phylum. Please, correct ssentence in order do not generate misunderstandings.
Ln224: Forsberg et al. [15]
Ln241: Chen et al. [78]
Maybe information under topic 6 (Ln265-295) can be extended a bit more. Same for topics 7, 8, 9.
Ln303: what probiotic? Please, be more specific.
Some of the studies cited were presented to generals, without appropriate details. Maybe authors can improve the presented sections with additional details, especially in topics discussed in sections 6, 7, 8, 9, 10.
Author Response
Reviewer 2
This is an interesting review paper, where authors have summarized the information on the relation between oral probiotics on the immunometabolic response during pregnancy and lactation. In my opinion paper deserve attention from the Editor of the journal and paper can be suggested for publication, however, revision, update and upgrade need to be considered as option for the improvement of the manuscript by the authors.
In my opinion abstract can be presented better where principal points later discussed in the paper can be summarized and presented.
Answer: Thank you for your mention. We rewrote the abstract. (Please see the attached manuscript.)
Ln46: Use of the word Lactobacillus is a quite delicate topic, since in 2020 the classification of former genus Lactobacillus was done. Well, it was suggested in same paper published in 2020 that to avoid misunderstanding between former (till 2020) genus Lactobacillus and new genus Lactobacillus (from 2020, as part of the reclassification), when referencing to works before 2020, English word "lactobacilli" can be applied. Moreover, in 2023, standardized abbreviations for the new genera derivate form former genus Lactobacillus were suggested. doi: 10.1099/ijsem.0.004107; doi: 10.1163/18762891-20230114
Answer: Thank you for your suggestion. We used the latest classification of the former genus Lactobacillus. (Please see the attached manuscript).
Ln65: maybe it will be better to add reference number [5] after Mor et al. in this position. Similar adjustments needs to be made in other parts of the manuscript.
Answer: We corrected it according to your recommendation. (Please see the attached manuscript).
Ln 75: Please, see my previous comment regarding use of term Lactobacillus / lactobacilli.
Answer: Thank you for your suggestion. We used the latest classification of the former genus Lactobacillus. (please see the attached manuscript).
Ln76: Citing genus Streptococcus and even Propionibacterium is quite delicate, since several species from that 2 genera are well known pathogens. Maybe some additional text needs to be added in this context and clarify the topic.
Answer: Thank you for your remark. We clarified the topic. (Please see the attached manuscript). Lines 80-83
In Table 1, Please, respect new names for lactobacilli and abbreviate them according to recommendations.
Answer: Thank you for your suggestion. We used the new names according to the latest classification of the former genus Lactobacillus. (Please see the attached manuscript).
Freudenreichii is one word; ssp. need to be subsp. then shermanii is not with capital S. Strain identification (GG, JS, etc.), do not need to be in italics.
Answer: Thank you for your mention. We corrected it. (Please see the attached manuscript).
Please, check table 1 and 2 and pay attention on abbreviation, use of ssp. that need to be changed to subsp.; use of coma and full stop, use of capital and not capital, etc. Simply review both tables.
Answer: Thank you for your remark. We review both tables. (Please see the attached manuscript).
Please, try to provide stain identification for all mentioned species in Table 1 and 2. This is important, since probiotic properties are strain specific and not species specific.
Answer: Thank you for your recommendation. We provided the strain identification. (Please see the attached manuscript).
Ln151-153: Since you have mentioned what is predominate in gut microbiota in pregnant women, then will be appropriate to mention how this is different to non-pregnant women. Are there some specific differences? This needs to be pointed out and appropriate references provided. Some points were clarified in the following 6 lines, but topic deserve a bit more attention.
Answer: Thank you for your valuable remark. We added new data. (Please see the attached manuscript). Lines 156-176
Ln160: Please, correct to Koren et al. [67].
Answer: I corrected it. (please see the attached manuscript).
In figure 2, please correct Lactobacillus to lactobacilli, and then abbreviate the mentioned species according to the recommendations.
Answer: Thank you for your suggestion. We fully wrote the mentioned species according to your recommendations. (Please see the attached manuscript).
Please, for the entire manuscript, after formally introducing specific microbial name, written in full, in following occasion, use abbreviation form. However, always when was mentioned for the first time, name needs to be written in full.
Answer: Thank you for your mention. We used the abbreviation form. (Please see the attached manuscript).
In the legend of Figure 2, all abbreviations used in the figure needs to be explained.
Answer: Thank you for your mention. We explained the abbreviation forms. (Please see the attached manuscript).
Ln170: Zhou et al. [69].
Answer: I corrected it. (Please see the attached manuscript).
Text into section Ln168-176 can be extended a bit more.
Answer: Thank you for your comment. We partially rewrote the section you mentioned. (Please see the attached manuscript).
Ln183: Klebsiella is not phylum. Please, correct sentence in order do not generate misunderstandings.
Answer: I corrected it. (Please see the attached manuscript).
Ln224: Forsberg et al. [15]
Answer: I corrected it. (Please see the attached manuscript).
Ln241: Chen et al. [78].
Answer: I corrected it. (Please see the attached manuscript).
Maybe information under topic 6 (Ln265-295) can be extended a bit more. Same for topics 7, 8, 9.
Answer: Thank you for your recommendations. We extended the sections you mentioned. (Please see the attached manuscript).
Ln303: what probiotic? Please, be more specific.
Answer: Thank you for your comment. We added more information. (Please see the attached manuscript).
Some of the studies cited were presented to generals, without appropriate details. Maybe authors can improve the presented sections with additional details, especially in topics discussed in sections 6, 7, 8, 9, 10.
Answer: Thank you for your recommendations. We inserted additional details in the sections you mentioned. (Please see the attached manuscript).
Kind regards
Reviewer 3 Report
Comments and Suggestions for Authors
The study conducts a systematic search in MEDLINE/PubMed to identify research that has examined the effect of probiotic interventions on the immunometabolic response during pregnancy and lactation, particularly in women with diabetes, overweight/obesity, preeclampsia, and allergic conditions. The process is appropriate: based on the database search, inclusion and exclusion criteria are established, articles are selected, and the results, discussion, and conclusions are presented. As potential improvements, it is suggested to reduce the manuscript length, especially by summarizing sections 4 to 11, and to clearly distinguish the results section, as it is not currently well defined and may be confused with the findings of the search process.
Author Response
Reviewer 3
The study conducts a systematic search in MEDLINE/PubMed to identify research that has examined the effect of probiotic interventions on the immunometabolic response during pregnancy and lactation, particularly in women with diabetes, overweight/obesity, preeclampsia, and allergic conditions. The process is appropriate: based on the database search, inclusion and exclusion criteria are established, articles are selected, and the results, discussion, and conclusions are presented. As potential improvements, it is suggested to reduce the manuscript length, especially by summarizing sections 4 to 11, and to clearly distinguish the results section, as it is not currently well defined and may be confused with the findings of the search process.
Answer: Thank you for your recommendations. We have rewritten certain parts and introduced additional details in the sections you mentioned to make the presentation of the information as clear as possible. (Please see the attached manuscript.)
Kind regards
Round 2
Reviewer 2 Report
Comments and Suggestions for Authors
nutrients-3578325-peer-review-v2
Authors have improved the text, however, quite neglecting the format of the manuscript. Finally, writing scientific papers needs to be according to the rules. More important is not to give wrong messages. When we discuss probiotic properties, we are referring to specific strains, since probiotic potential is not species characteristic, but is referring to the benefits from specific strain. Thus, strain identification needs to be provided for all cases. When you introduce for the first time specific scientific name in following occasions this name need to be abbreviated. Example: first time will be Lactiplantibacillus plantarum, and in next occasions will be Lpb. plantarum. This needs to be applied in entire text, including tables. When you have subspecies, “subsp” do not need to be in italics. Use of italics is to distinguish word that are not in English.
When you mention any microbial culture, on most occasions it needs to be with specific strain identification, since the probiotic properties are not species, but strain characteristics. In the manuscript, in most cases this information is missing. In answer to the comments was mentioned that this information was provided, however, in tables this information is missing. In case that you can not recover this information form the literature, then set a footnote and say, “strain identifications are not provided in the cited work”.
In Table 1, please, correct from “Rinne [9], 2005” to “Rinne [9]”. Apply this for all cited reference works in Table 1. Simply, check and adjust your manuscript to the Instructions for Authors and style recommended from the Publisher and Journal.
In the text consider correcting from “Lactobacillus” to “lactobacilli”. Please, check doi: 10.1099/ijsem.0.004107; doi: 10.1163/18762891-20230114. There was very well explained reasons for this. On Ln190, 341, etc. was corrected to lactobacilli, but this need to be without italics and without capital. Please, check the entire text and adjust. Please, ask a senior author of the manuscript to proof read the entire manuscript.
In the answers to questions, you have mentioned that corrections were made, however, in the manuscript this was not done.
Other example, Ln676-679: Streptococcus thermophilus, Lactobacillus acidophilus, Lacticaseibacillus paracasei (formerly Lactobacillus paracasei), Lactiplantibacillus plantarum (formerly Lactobacillus plantarum), Lactobacillus helveticus NCIMB 30440 (formerly Lactobacillus delbrueckii subsp. bulgaricus) and Bifidobacterium (breve, longum, lactis NCIMB 30436) in a dose of 112.5 ×109 and for 8 weeks is accompanied by a much better response [51]. Provide strain identification for cited strains; “Bifidobacterium (breve, longum, lactis NCIMB 30436”, NCIMB 30436 is from what of them? Please, check how to write names of the cultures; 112.5 x 109, correct to 1.13 x 1011. Again, simply ask some of more experienced colleagues to help you with formatting the manuscript.